# Genomic and Epidemiologic Surveillance of SARS-CoV-2 in the Pandemic Period: Sequencing Network of the Lazio Region, Italy

**DOI:** 10.3390/v15112192

**Published:** 2023-10-31

**Authors:** Martina Rueca, Giulia Berno, Alessandro Agresta, Martina Spaziante, Cesare Ernesto Maria Gruber, Lavinia Fabeni, Emanuela Giombini, Ornella Butera, Alessandra Barca, Paola Scognamiglio, Enrico Girardi, Fabrizio Maggi, Maria Beatrice Valli, Francesco Vairo

**Affiliations:** 1National Institute for Infectious Diseases ‘Lazzaro Spallanzani’, IRCCS, 00149 Rome, Italy; martina.rueca@inmi.it (M.R.); giulia.berno@inmi.it (G.B.); alessandro.agresta@inmi.it (A.A.); martina.spaziante@inmi.it (M.S.); lavinia.fabeni@inmi.it (L.F.); emanuela.giombini@inmi.it (E.G.); ornella.butera@inmi.it (O.B.); paola.scognamiglio@inmi.it (P.S.); enrico.girardi@inmi.it (E.G.); fabrizio.maggi@inmi.it (F.M.); mariabeatrice.valli@inmi.it (M.B.V.); francesco.vairo@inmi.it (F.V.); 2Direzione Regionale Salute E Integrazione Sociosanitaria, Area Promozione Della Salute E Prevenzione—Regione Lazio, 00145 Rome, Italy; abarca@regionelazio.it

**Keywords:** SARS-CoV-2, SARS-CoV-2 variants, genomic surveillance, epidemiology, Next Generation Sequencing, sequencing network

## Abstract

Since the beginning of the COVID-19 pandemic, large-scale genomic sequencing has immediately pointed out that SARS-CoV-2 has rapidly mutated during the course of the pandemic, resulting in the emergence of variants with a public health impact. In this context, strictly monitoring the circulating strains via NGS has proven to be crucial for the early identification of new emerging variants and the study of the genomic evolution and transmission of SARS-CoV-2. Following national and international guidelines, the Lazio region has created a sequencing laboratory network (WGSnet-Lazio) that works in synergy with the reference center for epidemiological surveillance (SERESMI) to monitor the circulation of SARS-CoV-2. Sequencing was carried out with the aims of characterizing outbreak transmission dynamics, performing the genomic analysis of viruses infecting specific categories of patients (i.e., immune-depressed, travelers, and people with severe symptoms) and randomly monitoring variant circulation. Here we report data emerging from sequencing activities carried out by WGSnet-Lazio (from February 2020 to October 2022) linked with epidemiological data to correlate the circulation of variants with the clinical and demographic characteristics of patients. The model of the sequencing network developed in the Lazio region proved to be a useful tool for SARS-CoV-2 surveillance and to support public health measures for epidemic containment.

## 1. Introduction

The coronavirus SARS-CoV-2 (Severe Acute Respiratory Syndrome Coronavirus 2) belongs to Coronaviridae family, subfamily Orthocoronavirinae, genus Betacoronavirus, subgenus Sarbecovirus 2, and is the causative agent of Coronavirus Disease-2019 (COVID-19) [1]. After its first detection in the city of Wuhan, Hubei province, China, at the end of 2019, the new virus spread rapidly in all parts of the globe, causing a pandemic [1], with more than 767 million confirmed cases and 6 million deaths [2].

The relevant evolutionary steps for the new virus to adapt to human populations are unknown; however, since the species jump, the single-strand positive-sense RNA genome of SARS-CoV-2 is continuously changing, with a mutation rate of 6.677 × 10^−4^ substitutions per site per year for the whole genome and 8.066 × 10^−4^ substitutions per site per year for the S gene [3]. Consequently, the primary concern was that genetic mutations accumulating over time could lead to the emergence of phenotypic variants with a greater impact on public health [4]. Indeed, the virus has constantly mutated throughout the pandemic and the original Wuhan strain has diverged into several SARS-CoV-2 lineages, which have been found circulating globally [5,6]. Moreover, several recombinant strains have been reported since coronaviruses naturally undergo a recombination process during viral replication.

The World Health Organization (WHO) has introduced a classification criterion for the different variants based on their proportion at the national and regional level and their impact on vaccines, therapies, and diagnostics, as well as the transmission and severity of the disease. Four groups of variants were defined: (1) Variants Of Concern (VOC); (2) Variants of Interest; (3) Variants of High Consequence; and (4) Variants Being Monitored [7,8,9,10]. The first SARS-CoV-2 variant was reported in the United Kingdom in 2020. It was named Alpha and soon showed increased transmissibility compared to the wild type, quickly becoming the most prevalent in Europe. Later, after May 2021, the circulation of the Alpha variant had drastically reduced in Europe due to the emergence and spreading of the Delta variant, which finally replaced Alpha.

At the end of 2021, a new variant, named Omicron, emerged in South Africa. It was classified as a VOC due to the high number of amino acid mutations detected, particularly in the S gene. After a few weeks of Delta and Omicron variants co-circulating, Omicron showed a significant growth advantage over Delta and higher transmissibility, leading to an increasing in the rate of incidence that had never been seen before in the pandemic. Since January 2022, the Omicron variant has become the predominant variant. However, it has diverged over time into many lineages and sub-lineages, which have been gradually identified and classified. The first was BA.1, then replaced by BA.2 and, although less sustained, by BA.3 and some recombinant strains (originating from Delta and Omicron variants) [11,12]. Afterwards, the identification of sub-variants BA.4 and BA.5 (mainly BA.5) raised concern due to their ability to evade host immunity as well as to be resistant to monoclonal antibody therapies, due to the presence of numerous mutations in the region coding for the Spike protein [13,14,15].

Consequently, it soon became apparent that the sequencing of the whole genome (WGS) of SARS-CoV-2 was the most effective method for studying the dynamics of virus transmission (i.e., epidemiological surveillance, contact tracing and outbreak investigation), the emergence of new mutations or variants potentially able to increase transmissibility and pathogenicity, or the impact on diagnostic methods. Moreover, WGS has proven to be a useful tool for the characterization of breakthrough infections or re-infections and for driving some therapeutic choices (i.e., the appropriateness of monoclonal antibody treatment).

The emergence of new variants has made the genomic surveillance of SARS-CoV-2 a priority globally and therefore international health authorities and the WHO have established a team of technical advisors to develop guidelines aimed at improving global sequencing capacity. Contextually, they called on countries to invest locally to establish an integrated high-performance genomic surveillance system including more structured sampling, WGS sequencing and data sharing [16,17,18]. In this context, the ECDC (European Center For Diseases Prevention and Control) recommended to European countries that they create networks for the surveillance of SARS-CoV-2 variants, based on interdisciplinary collaboration between national and local public health authorities, diagnostic laboratories and universities, to monitor the epidemiology of the circulating variants and detect the emergence of new strains early. Regarding laboratory activity, the suggestion was to build a network to coordinate sequencing activities and data analyses (including epidemiological data) with shared protocols. Indeed, SARS-CoV-2 is the first virus whose adaptive process to its new host has been monitored in real time since the first steps after the spillover through the analysis of the entire genome, thanks also to the sharing of genome sequences data on GISAID international databases and to the availability of the open-source data processing program Nextstrain [19,20].

Soon after the release of the ECDC recommendations [17], many countries created national platforms for sharing both genomic data and epidemiological metadata. Specifically, in Italy, a platform named I-Co-Gen (Italian-COVID19-Genomic) was developed, which can automatically analyze the raw sequencing data of the entire genome of SARS-CoV-2 as well as only the Spike gene; it is also equipped with an automatic alert system revealing variants of public health interest, whether new or already known [21].

In Italy, to monitor the national circulation of SARS-CoV-2, the Italian Ministry of Health established a network of regional reference laboratories for SARS-CoV-2 sequencing. In the region of Lazio, a WGS network (WGSnet-Lazio) including seven laboratories was established at the end of June 2021, coordinated by the Laboratory of Virology of the National Institute for Infectious Diseases “L. Spallanzani” (INMI) which is the regional reference laboratory. The WGSnet-Lazio performs sequencing weekly of the circulating strains collected according to the recommendations of the Ministry of Health. The results of the variants’ characterization are communicated to the LHUs (Local Health Units) and Regional Service for Epidemiology, Surveillance and Control of Infectious Diseases (SeRESMI), in order to allow for prompt implementation of adequate control and containment measures to prevent the diffusion of new potentially threatening SARS-CoV-2 variants.

In May 2020, a regional platform named Epidemia CoronaVirus (ECV) was developed for integrated COVID-19 virological and epidemiological surveillance, aiming to continuously and systematically collect information on all confirmed SARS-CoV-2 infections within the Lazio region. ECV proved to be a crucial tool in evaluating and monitoring the impact and evolution of the epidemic and it guided local authorities’ public health decisions and strategies (e.g., introducing or lifting non-pharmaceutical interventions). The aforementioned information was integrated with data obtained from SARS-CoV-2 sequencing provided by the WGSnet network and shared with public health regional authorities to monitor the epidemic, detect new strains in circulation early and drive public health strategies accordingly.

Here we report the results derived from these genomic surveillances of SARS-CoV-2 in the Lazio Region from February 2020 to October 2022.

## 2. Materials and Methods

### 2.1. Sampling Strategy

Since the emergence of SARS-CoV-2, the INMI virology laboratory has been producing sequences of the entire genome of SARS-CoV-2 by carrying out random sampling in parallel with targeted sequencing of particular cases. The number of samples selected had periodically increased, adapting to the recommendations indicated by international and national guidelines [17,18], until the involvement of other laboratories in the Lazio Region with NGS sequencing capabilities, culminating in the formation of the sequencing network WGSnet-Lazio, coordinated by our laboratory at INMI.

In particular, the workflow of the network included:I.Monthly flash surveys: the Istituto Superiore di Sanità (ISS) communicated the monthly collection day and the relative sample size, SeRESMI calculated the geographical distribution within the Lazio region and the INMI laboratory indicated to the diagnostic laboratories network (CoroNET-Lazio) the modalities and the number of samples requiring sequencing by one of the WGSnet laboratories;II.Continuous flow of weekly sequencing: in this case, the sampling, instead of being random, was targeted at hospitalized patients with severe COVID-19 disease and/or persistent SARS-CoV-2 infection and the total number of samples to be sequenced was based on ISS indications.

Several samples, mostly nasopharyngeal swabs, but also bronchoalveolar lavage (BAL) and sputum samples, were selected from the molecular tests performed with the real-time PCR method, which gave a result with a cycle threshold (Ct) less than 25.

All WGSnet partner laboratories contributed to sequencing activities according to a pre-established program. Samples collected were directly sent to the WGSnet laboratory with patient data attached, including the Ct value of the molecular test performed by the CoroNET laboratory that made the diagnosis.

The sequencing results had to be uploaded within 7 days on the I-Co-Gen platform that is managed by and directly connected with the ISS. In addition, the detected variants were returned to the local (LHU) and regional epidemiology service SeRESMI.

### 2.2. Sequencing

WGSnet laboratories used different commercial methods with an amplicon-based approach to obtain the whole genome of SARS-CoV-2 via NGS. Specifically, the “Ion Ampliseq SARS-CoV-2 Research Panel” kit, designed and commercially released by ThermoFisher (Waltham, MA, USA) in March 2020, was used for sequencing on the Ion Torrent platform. This method is based on the use of two pools of primers (total amplicons: 242; average length: 250 bps); the libraries were then sequenced on the GSS5 platform (ThermoFisher) to obtain approximately 250,000 reads/sample. The same primer panel was also used with the Genexus automatic sequencer (Ion Torrent; Thermofisher) and the Illumina Miseq platform by using V2 or V3 flowcells with the 2 × 150 paired-end method (AmpliSeq for Illumina SARS-CoV-2 Research Panel, Illumina). After this, a new version of the panel was developed to better amplify the S gene, which is the most mutated gene in Omicron compared to the previously circulating variants (SARS-CoV-2 Insight. Research Assay Panel—ThermoFisher).

WGSnet also used methods exclusively developed for Illumina platforms sequencing by several companies. Specifically, the assays most commonly used were NEBNext ARTIC SARS-CoV-2 Library Prep Kit (New England Biolabs, Ipswich, MA, USA) and COVIDSeq Assay (Illumina), which are based on amplification with the ARTIC primer set, designed and released by the ARTIC Network (last version: V.4) [22].

QIAseq DIRECT SARS-CoV-2 Kit (QIAGEN, Germantown, MD, USA) was also used, which is designed to generate 250 bp length amplicons without the need for fragmentation and ligation steps.

Sequencing steps were then performed on the Illumina platforms Miseq, Nextseq and Novaseq, according to manufacturer’s instructions (Illumina, San Diego, CA, USA). Further details regarding the SARS-CoV-2 WGS methods used by WGSnet-Lazio are available in Appendix A.

### 2.3. Sequence Analysis

The raw sequence data obtained were analyzed in order to generate a consensus sequence of SARS-CoV-2. The bioinformatics pipelines differ according to the method used for library preparation and the sequencing technique.

A lot of software has been developed during the pandemic for the reconstruction of the genomic sequence of SARS-CoV-2. Different tools are also used to obtain WGS by WGS-net laboratories, both home-made and commercial; most of those tools are characterized by freely available open-source code or by deriving data from available open-source platforms.

The home-made pipeline ESCA (Easy-to-use SARS-CoV-2 Assembler for Genome Sequencing) is a reference-based assembly algorithm, written for Linux environments and which requires only raw reads as input files, and has been designed to provide help to laboratories with low bioinformatic capacity, by using one single command. The algorithm has been used for analyzing Illumina paired-end reads in the “fastq.gz” file format, and Ion Torrent reads in the “ubam” file format. ESCA default setting parameters are specifically designed for amplicon-based protocols, its code is written in Linux and Python and is freely available from an open-source repository [23].

The assembly program IRMA (Iterative Refinement Meta-Assembler), developed by the CDC for the reconstruction of viral RNA genomes [24], is a meta-assembler that merges different components for viral genome analysis. IRMA provides read sorting, reference editing, variant calling and mutational phasing. The algorithm is written in Linux and Pearl and can be downloaded from the CDC portal.

The pipeline DRAGEN (Dynamics Read Analysis for GENomics) was designed for the analysis of reads produced with Illumina sequencers, and is freely available from the BaseSpace Illumina portal.

The WGS-net laboratories in Lazio deposited all genomic sequences into the national Integrated Rapid Infectious Disease Analysis—Advanced Research Infrastructure for Experimentation in GenomicS (IRIDA-ARIES) I-Co-Gen data portal provided by the Istituto Superiore di Sanità (ISS) [21]. Through I-Co-Gen data submission, the portal automatically performs a further whole-genome assembly through the built-in software RECoVERY [25], obtaining an WGS validation. Moreover, the IRIDA-ARIES portal allows us to rapidly share genomic and epidemiologic data between public health laboratories.

Finally, high-quality SARS-CoV-2 genomes have been directly submitted from I-Co-GEN to the GISAID data portal. All whole genome sequences deposited to GISAID have been used for further variants’ circulation and genomic characterization analysis.

### 2.4. Epidemiological Analysis

In May 2020, a regional platform (ECV) was developed for integrated COVID-19 microbiological and epidemiological surveillance, aiming to continuously and systematically collect information on all confirmed SARS-CoV-2 infections within the Lazio region.

ECV proved to be a crucial tool in evaluating and monitoring the impact and evolution of the epidemic and it guided local authorities’ public health decisions and strategies (e.g., introducing or lifting non-pharmaceutical interventions).

The platform was managed by the regional reference center (SERESMI); medical doctors from Hospital Health Directions and LHUs receiving case notifications from General Practitioners (GPs) and Primary Care Pediatricians (PCPs) were in charge of data entry, and they were asked to upload the demographical, clinical and epidemiological data of each laboratory-confirmed COVID-19 case. In particular, data collection was focused on the patient’s comorbidities, vaccination status, clinical course (e.g., disease severity, required hospitalization or intensive care, clinical outcome), attended setting/community (e.g., healthcare, school) and travel history.

At the same time, the ECV platform collected microbiological data from all regional reference laboratories across the Lazio region and, afterwards, data from all pharmacies allowed to perform antigenic or molecular SARS-CoV-2 testing as well.

Proportion for categorical variables and median with interquartile range for quantitative variables are reported. Chi-square and Wilcoxon–Mann–Whitney test were used to explore the presence of any statistically significant differences between the baseline demographic and clinical presentation of different SARS-CoV-2 variants. We used a logistic regression to analyze the relationship between our outcome, adjusting for variables associated with the baseline. The data were analyzed using STATA software (Version 17.0 SE—Standard Edition; Stata Corporation, College Station, TX, USA).

## 3. Results

### 3.1. The Circulation of SARS-CoV-2 Variants 

In the period between 28 December 2020 and 31 October 2022, the WGSnet released 12,418 sequences of whole genomes and/or the S gene of SARS-CoV-2, collected in the Lazio region. The graph in Figure 1 shows the total number of cases of SARS-CoV-2 infection reported weekly, from the last week of December 2020 to the end of October 2022 (green area), and the number of sequenced cases belonging to Alpha, Gamma, Delta and Omicron variants (colored lines). From the end of December 2020 to June 2021, the Alpha variant was predominantly in circulation; cases of infection with the Gamma variant (since February 2021) and Delta (since April 2021) were detected at a lower frequency. Since the last week of June 2021, the cases of infection caused by the Delta variant have increased rapidly, replacing Alpha variant cases, which disappeared in less than two months. The Delta variant continued to circulate with 100% prevalence up to the last week of November 2021, when the first cases of infection with the Omicron variant were reported in the Lazio region. The Omicron variant strongly increased from March 2022 until the last week of October 2022, accounting for all sequenced SARS-CoV-2 cases. The circulation of the Omicron sub-variants in the Lazio region highlighted that from March 2022 to June 2022 Omicron BA.2 was the most detected sub-variant; from June to July 2022 the major circulating sub-variant was BA.4, while from August 2022 to October 2022 the BA.5 sub-variant was detected in more than 90% of cases (data not shown).

The graph in Figure 1 also shows the trend of SARS-CoV-2 cases in the period from December 2020 to the end of October 2022. The trend of Omicron infection shows four peaks corresponding to the emergence of the Omicron sub-variants: BA.1, BA.2, BA.4 and BA.5, respectively. The first peak occurred during the first two weeks of January 2022 with 100,000 new cases/week, while the other peaks occurred from the last week of March to the first week of April 2022 and from the last week of June to the first of July 2022, respectively, with approximately 60,000 and 80,000 new cases/week. Of note, after the establishment of the WGSnet the number of sequences substantially increased (about 400 sequenced cases of the Delta variant despite a drop in cases). The sequences carried out remained above 250 until they increased again during the second half of November 2021, when more than 500 sequences were obtained per week. Starting from the second half of December 2021, Omicron accounted for all the sequences obtained and the sequencing number was proportional to the number of reported cases.

### 3.2. Circulation of Variants and Their Epidemiological Data

The median age of the infected patients was 47 (IQR 27–65) years. Patients infected with the Omicron variant were significantly older compared to the Alpha, Gamma or Delta patients’ groups (62, IQR 46–77 vs. 45, IQR 28–59, 44, IQR 25–57 and 36, IQR 20–53 years old, respectively, *p* < 0.001).

The graph in Figure 2 represents the distribution of the most circulating variants by age group in the Lazio region (Alpha, Delta and Omicron). It shows that the Delta variant accounted for more than one third (36.6%) of infections regardless of age group, mainly affecting the age groups under 40 years. The Omicron variant, on the contrary, accounted for most infections in older age groups, and was responsible for up to 60% of infections among patients over 60 years old.

Table 1 shows the baseline demographical and clinical features, stratified for SARS-CoV-2 variants. Patients from all variant arms were equally distributed among genders.

While Alpha variant infections were mostly symptomatic (59.7%), 68.1% of patients affected by Omicron variant infections were asymptomatic at the time of diagnosis. In Appendix A the clinical presentation of COVID-19 stratified for the variants is illustrated. Furthermore, this finding is confirmed via an analysis adjusted for age groups, which estimated that Omicron patients were more often asymptomatic at the moment of COVID-19 diagnosis compared to Alpha patients (OR 0.32; 95% CI 0.28–0.37).

Compared to patients infected with the Delta variant, patients infected with the Alpha or Omicron variant were more often affected by at least one comorbidity (respectively, 21.1% and 16.6% vs. 11.9% in the Delta group, *p* < 0.001). As expected, patients with the Omicron variant were more often fully vaccinated (78.5% versus 9%, 12.2% and 35.9% in the Alpha, Gamma and Delta groups, respectively, *p* < 0.001). Information on travel history and the place of infection (healthcare/family and relatives/other communities) were often missing during the Omicron period, due to the high incidence of new cases and pressure on the healthcare system, which had forcedly led to less accurate and less reliable contact-tracing activity.

### 3.3. SARS-CoV-2 Genomic Characterization

From 1 January 2020 to 31 October 2022, the WGSnet laboratories submitted 11,516 complete SARS-CoV-2 genomes to the GISAID database (the sequences set is provided in Appendix A).

Figure 3 shows the frequency of the major amino acid mutations found in the sequences of genomes submitted to the GISAID, both globally and by WGSnet. Overall, the frequency of mutations calculated in each of the two groups is comparable and most mutations have been detected in the S gene region. The D614G mutation has been found in almost 100% of sequences; about 60% of the Lazio region sequences (75% of the total GISAID sequences) present the T478Q; about 50% report the L452R and P681R mutations; and finally for above 20% the 69–70 deletion and the N501Y mutation is observed.

Additionally, there are various mutated sites in the N protein region (coding for nucleocapsid), in particular, the two R203K and G204R mutations were found with a frequency greater than 50% in all GISAID sequences and around 40% in the sequences from the Lazio region. However, the other coding regions are not affected by mutations except for the substitutions Q75Stop present in above 10% of GISAID sequences in the region of the NS8 protein and P323L in the NSP12 protein reported at a frequency close to 100% in all of the sequences.

The graph in Figure 4 represents the frequency of the same mutations analyzed in Figure 3 in the WGSnet sequences, stratified by variant (Alpha, Delta and Omicron). Omicron harbored most of the mutations with the majority of them located in the Spike protein region.

## 4. Discussion

This work reports the sequencing analysis of SARS-CoV-2 produced by the sequencing activities of the WGSnet network with the aim of monitoring variants circulating in the Lazio region, from the emergence of SARS-CoV-2 to the end of the pandemic period.

In the 2020–2022 period and after the identification of the first variant, Alpha, in early 2021, genomic surveillance activities have been systematically structured. In this regard, national and international guidelines [16,17,18] underline the crucial role of genomics monitoring, with the aim of early identification of the eventual emergence of variants with major fitness and/or pathogenicity.

Since the first weeks of January 2021, according to national and worldwide sequence data, the Alpha variant has been the most prevalent in the Lazio region (Figure 1) [26].

Alpha variant infections represented almost all the cases sequenced in the Lazio region until May 2021, when its circulation was replaced by the Delta variant. Awareness of the emergency and rapid diffusion of variants, with greater fitness and increased pathogenic potential, prompted international and national health authorities to strengthen genomic surveillance activities [16,17,18]. Following the publication of a ministerial ordinance, a sequencing network (WGSnet) was set up in the Lazio Region [27] and, consequently, the number of WGS sequencings rapidly increased, reaching peaks of 500 per week, as shown in Figure 1. Starting from the beginning of December 2021, a few days after the first identification of the Omicron variant and its rapid spread, a huge increase in cases was reported in the Lazio region, reaching up to 100,000 new cases weekly.

Over time, sequencing activities have increased progressively, and following international health authorities’ (i.e., ECDC, WHO) and national recommendation, from March 2022 the Lazio region has begun to sequence weekly, selecting cases with specific criteria, in parallel with random investigations [16,17,28]. This strategy provided two complementary monitoring approaches: one aimed at identifying variants in specific clinical categories or associated with a severe clinical course, and one aimed at characterizing circulating variants.

The present work demonstrates that the number of sequences performed was proportional to the trend of the cases reported in the Lazio Region. Indeed, constant genomic surveillance provided a strategy to strongly monitor circulating variants to identify the emergence of new strains with characteristics of potential concern for public health early.

Furthermore, analyzing patients’ demographic characteristics revealed an irregular distribution of viral variants among the different examined age groups. The Delta variant was more represented among younger patients (under the age of 50), and Omicron variant infections were most frequent in the older population (over 50 years old).

These data are possibly explained by the progression of the mass COVID-19 vaccination campaign conducted by national and regional health authorities (Appendix A). In fact, the Alpha variant predominantly circulated in the first months of 2021, when vaccination was mostly administered to older people, as shown in Appendix A. Viral circulation was mainly sustained by the Delta variant in the period between May and December 2021, when more than 60% of people over 60 years old had completed the vaccination cycle (Appendix A). At that time, only a minority of younger populations, mostly affected by this variant, were fully vaccinated (Appendix A). On the contrary, the Omicron variant began to affect the older age groups; these data are coherent with what emerged from several studies that documented a high potential for the Omicron variant to “escape” vaccine antibodies [26,29,30,31]. This is in line with the circulation trend of Omicron variant infections, which increased over time instead of most of the population being vaccinated (Appendix A).

Our data showed that the Alpha and Delta variants were usually associated with symptomatic infections whereas, in contrast, the Omicron variant mostly accounted for asymptomatic infections. This data is consistent with the literature: Omicron variants lead to mild infections with rare involvement of the lower airways and consequently a lower risk of hospitalization [31]. The benign course of Omicron variant infections could be due to two factors: the adaptation of the virus to the human host, and the protection conferred by vaccination against the severe clinical form of infection [32,33,34].

From the present study it also emerged that patients with the Omicron variant were frailer compared to those affected by the Delta variant. In fact, they were usually older and more often affected by at least one comorbidity; furthermore, they had more often received three vaccine doses compared to those in the Alpha, Gamma or Delta arms. This is consistent with the results that emerged from other studies [35] and may be possibly explained by the fact that frail patients more frequently undergo to COVID-19 testing, often in the context of hospital screening.

Comparing the amino acid mutations identified along all the genomes of SARS-CoV-2 in Lazio sequences with all sequences retrieved from the GISAID database reveals that the variability is comparable between the two groups, thus demonstrating that the circulation and evolution of the virus in the Lazio region mirrored global trends. Most amino acid substitutions are in the Spike protein and nucleo-capsid coding regions, and it is evident that Omicron is the variant that presents the greatest number of substitutions (Figure 4). Almost 100% of the sequences carry the D614G mutation (Spike protein) and P323L (NSP12, coding region for RdRp: RNA-dependent RNA polymerase). Both substitutions emerged in the first period of the pandemic and were held responsible for the increased transmissibility of the virus. The substitution D614G is linked with a higher affinity for the ACE2 cell receptor; P323L is associated with a greater tendency to introduce nucleotide mutations by RdRp, favoring the emergence of new mutated variants [36,37]. Furthermore, about 60% of sequences, and 100% of the Omicron sequences, (Figure 4) harbor the T478Q mutation in the region of the RBD, described as conferring increased replicative fitness [10,38]. About half of the sequences also have the mutations L452R and P681R (present in all Omicron variants, and in most Delta sequences, as shown in Figure 4). The first, localized in the RBD, is known to decrease the binding affinity of the monoclonal antibodies used for therapeutic purposes, reducing the efficiency of the drug [39]; the second instead confers increased replicative capacity and fusogenic power to the virus that favors the entry of the virus into the host cell [40]. The 69–70 deletion and the N501Y substitution (present in about 28% of the sequences), first highlighted in the Alpha variant, also emerged independently in some sub-lineages of the Delta variant and in 60 and 100% of the Omicron sequences, respectively. In particular, the amino acid position 501 is in the RBD region and is reported in the literature to positively influence the virus’ binding affinity with the receptor [38]. The amino acid substitution E484Q or E484K is also found in the RBD region; position 484 is mutated in about 6% of the sequences of the WGSnet and specifically in 100% of the Omicron sequences (Figure 3 and Figure 4). Such a replacement is known to give the variant the ability to “escape” from monoclonal antibody therapy and vaccine antibody neutralization [10,14].

## 5. Conclusions

Since the early stages of the COVID-19 pandemic, the role of genomic surveillance has been crucial to guide public health choices regarding the management of health measures and non-pharmaceutical countermeasures to contain the spread of the SARS-CoV-2 infection.

Transmission of SARS-CoV-2 infections has reached a great speed in the human population and the surveillance of circulating strains via WGS and the real-time global sharing of sequences have been crucial in understanding the dynamics of the spread and transmission of the infection.

The great efforts aimed at implementing genomic monitoring activity led to an increase in global sequencing capability. More efficient and less expensive methods were developed for obtaining the sequences of the whole viral genome; increasingly efficient and easy-to-use bioinformatics pipelines have been designed for analysis; new databases for wide sharing on national and international levels have been created and considerable investment has been made in the recruitment and training of laboratory personnel to implement the sequencing.

In Italy, a national sequencing network has been established and, similarly, the Lazio region, has created the WGSnet-Lazio network, which, in synergy with the diagnostic laboratories, has been working to monitor the strains circulating in the region.

Genomic surveillance based on sequencing of the whole genome integrated with data collected by the regional reference center (SERESMI) proved to be the best tool to obtain a clear and complete picture of the circulation of SARS-CoV-2. This integration of activities was able to detect the emergence of new variants, providing important information about the progress of the infection, the evolution of the virus, the response to therapies and the efficiency of the prevention measures.

The model of the WGSnet-Lazio network, and the increased capability developed within the monitoring of SARS-CoV-2 infections, will be able to continue to support the surveillance of circulating strains of SARS-CoV-2 and to provide the basis for the development of future network programs for new emerging or re-emerging pathogens.

## Figures and Tables

**Figure 1 viruses-15-02192-f001:**
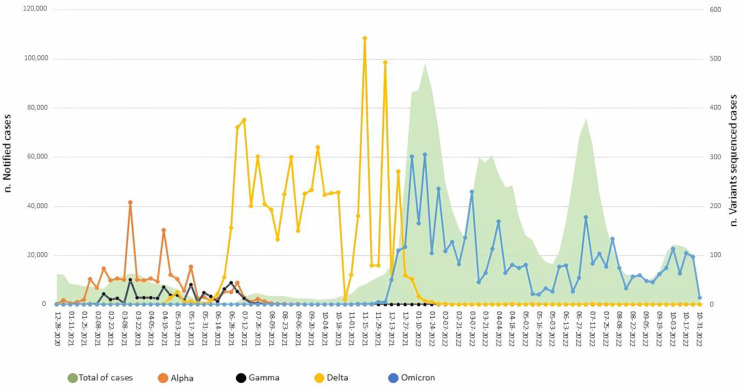
Reported cases of SARS-CoV-2 infection and sequenced cases of variants that most circulated in the Lazio region in the period between 28 December 2020 and 31 October 2022. The green area represents the number of cases reported per week (numerical scale on the left-hand axis); the lines represent the number of sequenced cases with variant infection caused by Alpha (orange line), Gamma (black line), Delta (yellow line) and Omicron variants (blue line), with the numerical scale indicated on the right-hand axis.

**Figure 2 viruses-15-02192-f002:**
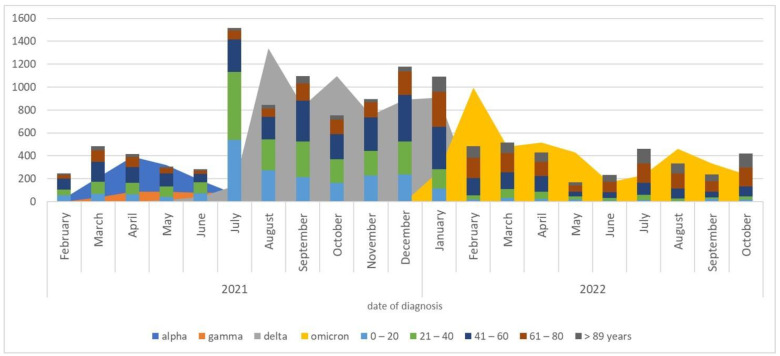
Distribution of infections caused by Alpha, Delta and Omicron variants by age group.

**Figure 3 viruses-15-02192-f003:**
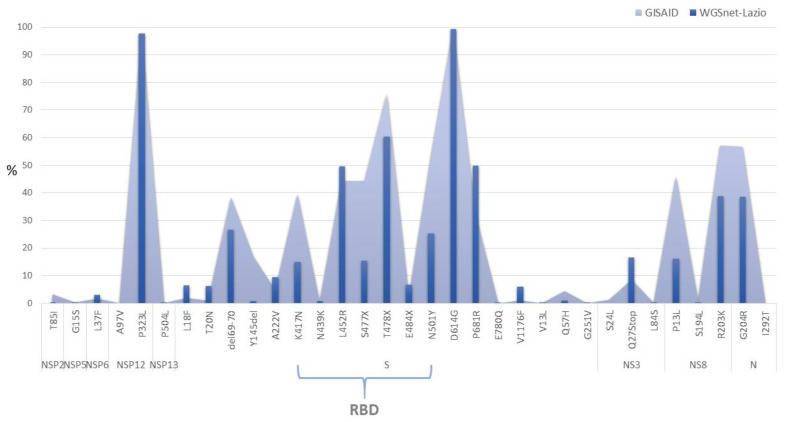
Frequency of amino acid mutations most found in sequences present in the GISAID database. The dark blue bars represent the frequency of mutations in WGSnet sequences, the celestial area represents the frequencies in all of the sequences present on the GISAID platform globally. RBD: Receptor Binding Domain.

**Figure 4 viruses-15-02192-f004:**
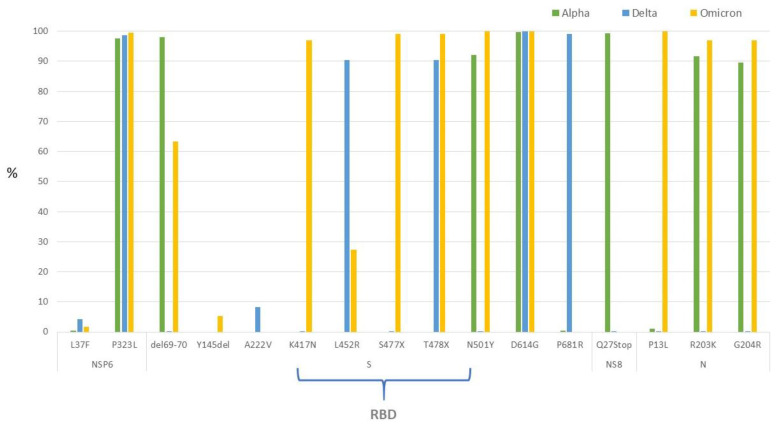
Frequency of amino acid mutations most found in sequences submitted by WGSnet-Lazio to the GISAID platform for the Alpha, Delta and Omicron variants present in the GISAID database. The yellow bar represents the frequency of mutations in Alpha sequences, the blue one the frequency of mutations in the Delta sequences and the green one the frequency of mutations in Omicron sequences. RBD: Receptor Binding Domain.

**Table 1 viruses-15-02192-t001:** Baseline demographical and clinical features stratified for SARS-CoV-2 variants. After adjustment for age group, presence of comorbidities and vaccination status, patients infected with the Alpha variant had a higher mortality rate compared to the Delta patients (OR 2.80 95% CI 1.73–4.54 *p* < 0.001).

	Alpha	Gamma	Delta	Omicron	Total	
	(N = 1302)	(N = 460)	(N = 6113)	(N = 4543)	(N = 12418)	*p* Value
Age, median (IQR)	45, IQR (28–59)	44, IQR (25–57)	36, IQR (20–53)	62, IQR (46–77)	47, IQR (27–65)	0.000
Gender, n (%)						
Female	643 (49.4)	232 (50.4)	2972 (48.6)	2262 (49.8)	6109 (49.2)	
Male	659 (50.6)	228 (49.6)	3141 (51.4)	2281 (49.2)	6309 (50.8)	0.626
Age groups in years, n (%)						
0–20	223 (17.1)	81 (17.6)	1648 (27.0)	299 (6.6)	2251 (18.1)	
21–40	330 (25.4)	125 (27.2)	178 (29.1)	586 (12.9)	2821 (22.7)	
41–60	445 (34.2)	163 (35.4)	1702 (27.8)	1278 (28.1)	3588 (28.9)	
61–80	236 (18.1)	62 (13.5)	759 (12.4)	1494 (39.9)	2551 (20.5)	
>80	68 (5.2)	29 (6.3)	224 (3.7)	886 (19.5)	1207 (9.7)	0.000
Symptoms at diagnosis, n (%)						
Yes	777 (59.7)	309 (67.2)	3242 (53.0)	1451 (31.9)	5779 (46.5)	
No	525 (40.3)	151 (32.8)	2871 (47.0)	3092 (68.1)	6639 (53.5)	0.000
Comorbidities, n (%)						
Yes	275 (21.1)	91 (19.8)	727 (11.9)	755 (16.6)	1848 (14.9)	
No	537 (41.2)	210 (45.7)	2038 (33.3)	213 (4.7)	2998 (24.1)	
Missing data	490 (37.6)	159 (34.6)	3348 (54.8)	3575 (78.7)	7572 (61)	0.000
Travel, n (%)						
Yes	60 (4.6)	24 (5.2)	491 (8)	42 (0.9)	617 (5)	
No	813 (62.4)	284 (61.7)	2498 (40.9)	802 (17.7)	4397 (35.4)	
Missing data	429 (32.9)	152 (33)	3124 (51.1)	3699 (81.4)	7404 (59.6)	0.000
Infection setting, n (%)						
Healthcare setting	32 (2.5)	23 (5)	60 (0.9)	51 (1.2)	166 (1.3)	
Household/friends	414 (31.8)	156 (33.9)	1156 (18.9)	141 (3.1)	1867 (15)	
Other settings	196 (15.1)	58 (12.6)	631 (10.3)	70 (1.5)	955 (7.7)	0.000
Missing data	660 (50.7)	223 (48.5)	4266 (69.8)	4281 (94.2)	9430 (75.9)	
Vaccination status, n (%)						
Not vaccinated	1075 (82.6)	334 (72.6)	3428 (56.1)	878 (19.3)	5715 (46)	
Uncomplete vaccination	110 (8.4)	70 (15.2)	491 (8)	101 (2.2)	772 (6.2)	
Fully vaccinated	117 (9)	56 (12.2)	2194 (35.9)	3564 (78.5)	5931 (47.8)	0.000

## Data Availability

The dataset of SARS-CoV-2 complete genome sequences submitted by WGSnet laboratories to the GISAID database is available as Appendix A. De-identified data that underlie the results reported in this article will be made available upon reasonable request. Proposals should be directed to the corresponding author.

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
