# Peer review of "Genomic and Epidemiologic Surveillance of SARS-CoV-2 in the Pandemic Period: Sequencing Network of the Lazio Region, Italy"

_viruses, 2023, doi:10.3390/v15112192_

Round 1

Reviewer 1 Report

Comments and Suggestions for Authors

The manuscript “Genomic and epidemiologic surveillance of SARS-CoV-2 in the pandemic period: sequencing network of Lazio Region, Italy” by Rueca et al. analyzes the epidemiology and variant dynamics in the Lazio region between December 2020 and October 2022. The authors make use of a powerful database built on a network of Italian laboratories and researchers to generate results.

My main concern is that the results are mainly confirmation of research already conducted on SARS-CoV-2 in the past years around the world. Novelty could be inserted by delving deeper into the epidemiological data the authors have access to. Based on the current manuscript alone, I believe the “full article” format may be too extensive and it should be reduced to fit the report format. I have some questions that I think should be addressed prior to acceptance.

1)      Abstract, line 18: Authors point out that SARS-CoV-2 is constantly evolving, but all life is. Maybe “rapidly evolving”? Or they meant that genomic surveillance is capable of monitoring this evolution?

2)      Length of paragraphs in the introduction should be adjusted to improve readability. It is too long as some of the information provided is too general to matter in understanding the manuscript. E.g. the 8th paragraph (lines 105-110) should be combined into a single phrase.

3)      Line 67: What do the authors mean by “Since January 2022, the Omicron variant has been permanently fixed”? If the authors meant “fixed” as in “a haplotype fixed in the population”, there are sequences classified as being Delta in the GISAID database that were collected in 2023 (If the dates are correct).

4)      Line 75: The authors state “as the evolutionary processes of Omicron continued during the pandemic” As in the abstract, this could be interpreted as continued evolution not being the norm in all life, including viruses.

5)      I also feel that by the end of the introduction the readers don’t know what the objectives of the investigation are nor the main results expected.

6)      Through the “Methods” section,  the authors state the methods and pipelines used in a general way. This decreases the reproducibility of the paper. I understand that different laboratories may have different protocols and their full description be too long for the main text of the manuscript, but they could be made available as a supplementary file. Likewise, If homemade pipelines are too dense to be described in the text, the scripts should be provided.

7)      The section “2.4. Epidemiological analyses” should be tailored to directly describe the data or methods used, instead of a more general description of the ECV platform. This description can be moved to the introduction.

8)      I don’t think the best way to represent the data is the one provided in Figure 2. As each variant was most present at a different time, surges in cases can skew the proportions seen in the figure. E.G: The authors discuss that Alpha had similar infection rates in all age groups, but the literature indicates otherwise (eg. Ladhani et al 2021, Lindstrøma et al 2021, Loconsole et al, 2021). In this figure, I would prefer the bars to be months and the colors for the age groups. Each variant could be represented by a timeframe. Also, the text size is too small for the figure.

9)      Figure 3 is a little bit redundant. If 60% of alpha cases are symptomatic, the remaining 40% are assumed to be asymptomatic. Figure should be excluded and its information fully incorporated into the text.

10)   I miss more exploration of the epidemiological results. As an example, are the asymptomatic cases in each variant more related to one age range than the other? The authors also stated in the methods that the database used has information on “patient’s comorbidities, vaccination status, clinical course (e.g., disease severity, required hospitalization or intensive care, clinical outcome), attended setting/community (e.g., healthcare, school) and travel history”. Why were these not analyzed? The epidemiological analyses could bring more novelty to the paper, as variant dynamics and mutation frequencies have already been extensively analyzed in the literature.

11)   In Figure 5, it seems the total frequencies of each mutation are scaled to 100%. Because of this, a barplot would be better suited to represent the data than the format presented. To increase readability, mutations with 0% representation (as seen in Figure 4) should be removed from Figure 4.

12)   Text size in Figures 4 and 5 should be increased.

Comments on the Quality of English Language

English is good, although a review by the authors might improve it. More than that, I would recommend a review of the text so that it's not needlessly verbose.

Reviewer 2 Report

Comments and Suggestions for Authors

This manuscript entitled “Genomic and epidemiologic surveillance of SARS-CoV-2 in the pandemic period: sequencing network of Lazio Region, Italy” aimed to describe a SARS-CoV-2 genomic surveillance network in the Lazio Region, Italy, and variants and epidemiology results from Dec. 2020 to Oct. 2022. The manuscript is well written in introduction and methodologies, however, the analysis and reported results are too simple to reflect the whole surveillance effort of such a long time; as a result, the meaning of the manuscript seems not well represented. More analysis of variants and collected host data would be needed to strengthen the meaning of genomic and epidemiologic surveillance in the Lazio Region. Also, citations need to be updated.

Please see below my line-to-line comments.

Lines 180-184: Please provide more details of the bioinformatics pipelines. 

Line 190: Please provide the full name of the regional platform ECV.

Line 231: Why there’s a discrepancy in the line and the green area in the delta time in Figure 1? 

Line 237-241: What are the proportions of the previously mentioned “monthly flush survey” and “continuous flow of weekly samples”? Is this totally random sampling? How can the author be certain that the sequenced data is reflective of variants from all the cases circulating out there?

Line 250: Are patient numbers in each age group similar? Why only the patient's age was chosen for this analysis? In lines 200-203, it said “comorbidities, vaccination status, clinical course (e.g., disease severity, required hospitalization or intensive care, clinical outcome), attended setting/community (e.g., healthcare, school) and travel history” were the focus of the data collection; but why no analysis on these collected data? Analysis on age only could not reflect of good picture of the epidemiology data analysis. 

Line 257: Figure 3 can be in supplemental.

Line 268: This section simply compared some mutations observed in the author’s dataset with those on GISAID. But again, without patients' metadata input, the overall value of such analysis seems to be low.

Lines 269-276: These should be in methods. Actually, an additional methods section of data submission may be needed.

Please also consider revising Figure 5. The area graph is very misleading while only the frequency is needed.

Comments on the Quality of English Language

English writing is fine.

Round 2

Reviewer 1 Report

Comments and Suggestions for Authors

The manuscript has been greatly improved with the new additions and changes to the text. Some small points still need attention from the authors:

a) The introduction is still too long and a more objective/succinct writing would help. The paragraphs contained in lines 96-127 are a little bit confusing to somebody not familiar with the Italian networl of SARS-CoV-2 surveillance. Maybe the relationship between the Italian Ministry of Health, I-Co-Gen, IRIDA-ARIES, ISS, INMI, CoroNET-Lazio, WGSnet-Lazio, LHUs, SeRESMI and ECV could be summarized in a fluxogram? I could understand better the dynamics between these by the description contained in the “Sampling Strategy” (itens I and II) section of the Methods than by the information contained in the introduction.

b) The only supplementary file available to me was Table S1. Figures S1 and S2 and Table S2 should also be provided.  Also, Table S2 is not referenced in the text.

c) In line 395, the authors state that “The Alpha variant caused on average 30% of infections in all age groups”, but that’s not what it is shown in Figure 2.

Small suggestions:

·       Paragraphs 2, 3 and 4 (lines 42-53) should be formatted into a single paragraph.
·       Paragraph 5 should be divided into two, at line 63.
·       Paragraph 6 should be divided into two, at line 82.
·       The ECDC acronym (first appearing at line 89) has not been defined in the text.
·       Modify “As regards” to “Regarding” in line 93
·       Acronyms that are used only one or two times in the text should be removed (e.g. VOI, VOC, VOHC, VBM, LHU)
·       Modify “as well as the only full-length of the Spike gene” to “as well as only the Spike gene”
·       On the x-axis of Figure 2, it is written Gennaio instead of January.
·       Include a comma after “(59.7%)” in line 308

Comments on the Quality of English Language

No further comments

Reviewer 2 Report

Comments and Suggestions for Authors

There is a typo in Table 1: travel history. The manuscript is well-improved.

Comments on the Quality of English Language

The English writing is fine.
